# IL17 factors are early regulators in the gut epithelium during inflammatory response to *Vibrio* in the sea urchin larva

Katherine M Buckley[1,2*†‡], Eric Chun Hei Ho[2,3†], Taku Hibino[2§], Catherine S Schrankel[1,2], Nicholas W Schuh[2,3], Guizhi Wang[2], Jonathan P Rast[1,2,3*]

[1]Department of Immunology, University of Toronto, Toronto, Canada; [2]Sunnybrook Research Institute, Toronto, Canada; [3]Department of Medical Biophysics, University of Toronto, Toronto, Canada

**Abstract** IL17 cytokines are central mediators of mammalian immunity. In vertebrates, these factors derive from diverse cellular sources. Sea urchins share a molecular heritage with chordates that includes the IL17 system. Here, we characterize the role of epithelial expression of IL17 in the larval gut-associated immune response. The purple sea urchin genome encodes 10 IL17 subfamilies (35 genes) and 2 IL17 receptors. Most of these subfamilies are conserved throughout echinoderms. Two IL17 subfamilies are sequentially strongly upregulated and attenuated in the gut epithelium in response to bacterial disturbance. IL17R1 signal perturbation results in reduced expression of several response genes including an IL17 subtype, indicating a potential feedback. A third IL17 subfamily is activated in adult immune cells indicating that expression in immune cells and epithelia is divided among families. The larva provides a tractable model to investigate the regulation and consequences of gut epithelial IL17 expression across the organism.

*For correspondence: kshank@ email.gwu.edu (KMB); jrast@sri. utoronto.ca (JPR)

†These authors contributed equally to this work

Present address: ‡The George Washington University, Washington, United States; §Faculty of Education, Saitama University, Saitama, Japan

**Competing interests:** The authors declare that no competing interests exist.

## Introduction

Gut epithelial cells deploy an elaborate suite of signals to transmit information about the state of the gut lumen to the wider organism. These communication networks can be difficult to interpret in the context of vertebrate systems, which exhibit complexity at both morphological (e.g. vertebrate guts are multilayered tissues that interact with many types of peripheral immune cells) and molecular levels. The gut is an ancient site of intense immune activity, and core aspects of the regulatory circuitry in this tissue are likely to be conserved across phyla. Consequently, invertebrate animals provide alternative models to investigate the fundamental mechanisms that control the connections between the gut lumen and the distributed immune system. Some of these organisms are morphologically and genetically simple, which provides unique experimental advantages, including reduced microbiota diversity, optical transparency and efficient transgenesis.

The difficulty in identifying homologs of mammalian cytokines, even within other vertebrate classes (*Secombes et al., 2011*), remains a long-standing barrier to this approach. As central mediators of the immune response, cytokines are key targets for pathogen mimicry or co-option (*Elde and Malik, 2009*; *Epperson et al., 2012*) and are subject to high levels of evolutionary pressure and sequence diversification (*Koyanagi et al., 2010*). An exception to this trend is the IL17 cytokine family. These proteins are characterized by a cysteine-knot fold structure that is formed through interactions among four conserved cysteine residues (*Hymowitz et al., 2001*). This structural constraint provides a means to computationally identify IL17 homologs across phyla. IL17 cytokines have been functionally characterized in jawed (*Kolls and Lindén, 2004*) and jawless vertebrates (*Smith et al., 2013*; *Han et al., 2015*), and orthologs have been identified also in invertebrate deuterostomes

**eLife digest** To protect themselves from the constant invasion of harmful microbes, animals have evolved complex immune systems. The gut is one of the most active sites of the immune system and plays a key role in regulating immune responses. In mammals, cells lining the gut wall can sense the presence of harmful bacteria and communicate this information to tissues across the body by producing specialized proteins called Interleukin-17 (IL-17). IL-17 proteins are important for regulating inflammation and are thought to activate specific immune cells in an infected area.

Some aspects of immune systems are similar between different animal species, which can provide clues of how immunity evolved and how it is regulated. For example, sea urchins, which evolved 400-600 million years ago, begin life as simple larvae consisting of a few thousand cells. As oceans harbor a multitude of bacteria and viruses, sea urchin larvae need an efficient immune system to defend themselves. These larvae can respond to specific types of bacteria within a few hours after the microbes have entered their gut by modifying gene expression in distant cells. As these changes occur in cells that are removed from the bacteria, it is thought that the gut cells that initially sense the bacteria, somehow communicate this information.

Now, Buckley et al. exposed sea urchin larvae to a marine bacterium and measured the responses of the cells and their gene expression. The infection affected several types of cells, and in the first 24 hours, a subset of immune cells changed shape and started migrating to the gut wall. In addition, IL-17 gene expression changed significantly in gut cells in the early phases of the larval immune response. Buckley et al. identified three types of IL-17 proteins involved in sea urchin immunity: two that are important for the immune response in the gut during the larval stage, and a third that is only present in adults. These findings suggest that IL-17 signaling is an ancient and central element of gut-associated immune response, which even exists in animals that evolved long before humans.

These findings demonstrate that the sea urchin larva represents a unique and ideal experimental model to study immune responses in a living organism that is more closely related to mammals than some other models, like fruit flies or worms. By understanding the fundamental mechanisms that mediate gut health, this work may highlight new drug targets to treat conditions like Crohn's disease and colon cancer.

(*Huang et al., 2008*; *Hibino et al., 2006*) molluscs, nematodes and arthropods (*Daphnia*) (*Huang et al., 2015*). In contrast, IL17 factors are absent from available genome sequences of insects and non-bilaterian metazoans. The broad phylogenetic distribution of this signaling system underscores the fundamental role of the IL17 family in animal biology and highlights the opportunity to glean understanding of this system using experimentally accessible invertebrate models.

Most mammalian genomes encode six IL17 family members (IL17A-F) (*Kolls and Lindén, 2004*), of which the most widely studied are the closely related IL17A and IL17F. These two highly expressed cytokines define subsets of effector T cells (Th17 cells and γδ17 cells) and innate lymphocyte-like cells (ILCs) and induce strong inflammatory responses (*Korn et al., 2009*; *Lockhart et al., 2006*; *Gladiator et al., 2013*). Importantly, IL17 expression is not restricted to lymphocytes or other mesodermal immune cells. Three members of the IL17 family (IL17B, IL17C, and IL17E [also known as IL-25]) are expressed by epithelial cells, including those in the gut (*Song et al., 2011*; *Ramirez-Carrozzi et al., 2011*; *Reynolds et al., 2015*). In this context, IL17C is a key factor in the early intestinal immune response where it regulates the expression of many innate immune genes. In colonic epithelial cells, IL17E promotes inflammation through the IL17RB receptor, while IL17B competitively binds with IL17RB to interfere with this signal (*Reynolds et al., 2015*). Thus, within humans and mice, the IL17 cytokines are produced from a variety of cellular sources and have wide-ranging functions and downstream transcriptional consequences, some of which are just beginning to be understood.

To investigate the role of IL17 in the gut-associated immune response within the context of a morphologically simple organism, the larval stage of the purple sea urchin (*Strongylocentrotus purpuratus*) provides a unique model system (*Ch Ho et al., 2016*). The purple sea urchin genome

sequence encodes an expansive set of immune receptors and effectors as well as conserved signaling pathways downstream of pattern recognition receptors and homologs of a suite of transcription factors that have key roles in modulating the immune response and hematopoiesis in vertebrates (*Hibino et al., 2006*; *Sodergren et al., 2006*; *Rast et al., 2006*; *Messier-Solek et al., 2010*; *Buckley and Rast, 2012*, *2015*; *Solek et al., 2013*; *Schrankel et al., 2016*). This genetic heritage is shared within the deuterostomes (e.g. echinoderms, hemichordates and chordates), providing a context in which to investigate IL17 function in a simple invertebrate for comparison to mammals.

Purple sea urchins undergo indirect development with a bilaterally symmetric, planktonic larval form that metamorphoses into a pentameral adult. Over 5 days, embryos synchronously develop to form free-swimming larvae that feed for 10–12 weeks before settling and metamorphosis (reviewed in [*McClay, 2011*]). At 10 days post-fertilization (dpf), larvae are 300–400 µM in length and consist of about 4000 cells. Larvae have a tripartite gut composed of an epithelial monolayer separated by two sphincters into a pharynx, midgut and hindgut (*Smith et al., 2008*). Larvae have several types of immune cells (*Ch Ho et al., 2016*) including a granular cell population known as 'pigment cells' and a heterogeneous suite of several types of 'blastocoelar cells' that populate the body cavity (blastocoel) (*Solek et al., 2013*; *Tamboline and Burke, 1992*; *Gibson and Burke, 1985*). Collectively, these cells mediate the larval immune response through surveillance-like motility, phagocytosis, expression of immune effectors and regulatory cell-cell interactions (*Ch Ho et al., 2016*). The simplicity and optical transparency of the sea urchin larva allows visualization and quantification of the immune response on an organism-wide scale at single-cell resolution.

We have developed a model for gut-associated immune response in which larvae are exposed to the Gram-negative bacterium *Vibrio diazotrophicus* (*Ch Ho et al., 2016*). This marine bacterium was first isolated from the gastrointestinal tract of the congeneric green sea urchin, *S. droebachiensis* (*Guerinot et al., 1982*). Other *Vibrio* species have been implicated as causative disease agents in several adult sea urchins species (*Becker et al., 2008*). Upon exposure to *V. diazotrophicus*, larvae exhibit a synchronous and robust set of cellular responses over a period of 24 hr (*Ch Ho et al., 2016*). The most notable of these is that a subset of pigment cells change shape from a stellate to a rounded form, disengage from their typical positions apposed to the aboral ectoderm, and migrate to the gut epithelium. Some blastocoelar cell types exhibit changes in cell motility and increasingly frequent cell-cell interactions with each other and with the gut epithelium. Additionally, gut morphology is affected: the epithelial wall thickens to constrict the midgut, suggesting that the animals cease feeding. By 24 hr, bacteria are evident within the blastocoel, where they are phagocytosed by filopodial blastocoelar cells. Early changes in gene activity are most evident within the gut epithelium within 2 hr of exposure, which is well before bacteria penetrate the gut lumen. Transcriptional affects are also apparent in peripheral immune cells. These observations suggest that the system-wide response is regulated by recognition of microbial disturbance at the gut epithelium and is mediated in part by signals that transmit the state of the gut lumen to cells distributed throughout the organism.

Through comprehensive surveys of larval gene activity during infection, we find that two small subfamilies of *IL17* genes emerge as highly regulated factors during the early response. Here, we address the genomic repertoire, expression and function of the IL17 cytokines and receptors in the purple sea urchin immune response. We also present the diversity of IL17 sequences within the purple sea urchin genome with reference to other echinoderms. Expression of the sea urchin *IL17* genes is evident only after bacterial exposure and is restricted to the gut epithelium in this infection model, as assessed by both *in situ* hybridization and transgenic reporters. In the larva, exposure to *V. diazotrophicus* does not elicit expression of IL17 in mesodermally derived immune cells. In contrast, a third subfamily of *IL17* is acutely expressed in the adult by circulating immunocytes in response to immune challenge and injury. The parallel roles of these IL17 subfamilies within the sea urchin immune response mirror the similar division of labor among vertebrate IL17 factors and highlight fundamental aspects of animal immunity. Functional data in the larva indicate that disruption of IL17 signaling leads to decreased expression of several immune regulators and effector genes in the gut epithelium, including some of the IL17 factors. Collectively, these findings indicate that epithelial expression of IL17 family regulators is central to an ancient aspect of gut immunity

# Results

## A genome-wide survey identifies IL17 factors as an acutely upregulated signal in immune response

Seawater exposure to the marine bacterium *V. diazotrophicus* (*V.d.*) induces a distinct cellular response in sea urchin larvae that includes the migration of pigment cells to the gut epithelium, changes in cell behavior and altered gut morphology (*Figure 1a,b*) (*Ch Ho et al., 2016*). To investigate the transcriptional underpinnings of this response, whole transcriptome sequencing was performed on mRNA isolated from larval samples collected at 0, 6, 12 and 24 hr of exposure to *V.d.* Given the morphological simplicity of the sea urchin larva and the depth of sequence coverage, these data provide a system-wide picture of biologically relevant transcriptional state changes upon bacterial exposure.

The sea urchin genome encodes a large complement of genes with homologs involved in immunity in other organisms, including pattern recognition receptors, signaling molecules, immune effector and transcription factors (*Hibino et al., 2006*). Analyses of our RNA-Seq screens indicate that

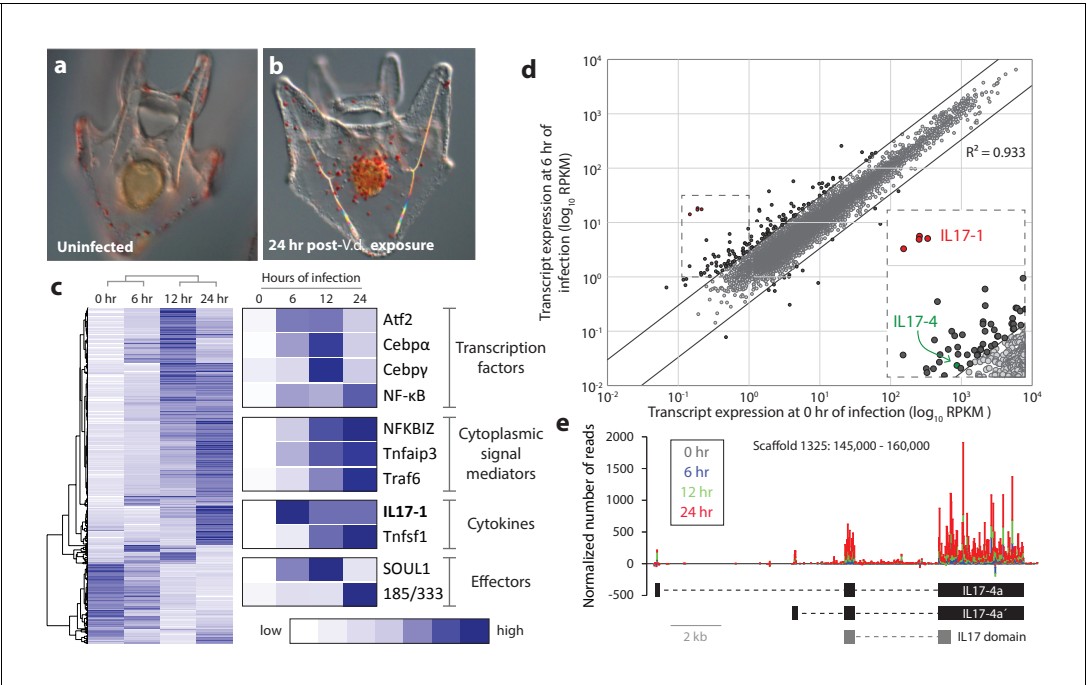

**Figure 1.** Sea urchin larvae exhibit changes in cell behavior and gene expression following exposure to specific bacterial strains. (a,b) The larval cellular immune response includes pigment cell migration to the gut epithelium. An uninfected control larva (a), and a larva exposed to *V.d.* for 24 hr (b) are shown. The red color of pigment cells is a consequence of echinochrome A, a naphthoquinone that is encapsulated in large granules. Under typical laboratory conditions, pigment cells localize to the outer ectoderm. In response to certain bacterial isolates, these cells migrate through the blastocoel to interact with the gut epithelium (*Ch Ho et al., 2016*). (c) Genes are activated with a variety of kinetics that varies among functional classes. RNA-Seq was performed on larvae collected at 0, 6, 12, and 24 hr of exposure to *V.d.* Differentially expressed transcripts (RPKM ≥3, fold-change in expression ≥2; 3238 total transcripts) were hierarchically clustered with average linkages to identify similarly temporally regulated genes using Gene Cluster 3.0. High expression is indicated in dark blue; low expression is shown in white. (d) A subset of IL17 genes is upregulated early in infection. Transcript levels are shown for 16,920 genes that are expressed in larvae during the infection time course. Expression levels (RPKM) are shown for uninfected larvae (x-axis) and larvae exposed to *V.d.* for 6 hr (y-axis). Most genes were not strongly differentially expressed at these two time points (gray). The 127 genes that exhibit ≥3-fold changes in expression levels between 0 and 6 hr of infection are shown in black. The genes within the *SpIL17* subfamilies are indicated (*SpIL17-1*, red; *SpIL17-4*, green). The dashed box is enlarged in the inset. (e) Mapped reads were used to identify a novel *SpIL17* transcript. RNA-Seq reads were mapped to the *S. purpuratus* genome. Genomic regions that contained mapped reads but no known genes were investigated for domains that are associated with immunity (Scaffold1325:145,000–160,000 from *S. purpuratus* genome, v3.1 is shown). The number of reads normalized to library size is plotted for each time point as a stacked bar graph. Reads that map to the positive strand are shown with positive values; negative values indicate reads that map on the negative strand. The exons of the experimentally confirmed *SpIL17-4a* transcript sequences (*SpIL17-4a* and *SpIL17-4a'*) are indicated in black. The location of the IL17 domain encoded by the transcripts is shown in gray.

much of this genetic complexity is deployed within the larval immune response (*Figure 1c*). This includes the expression of homologs of important immune transcription regulators in vertebrates (e. g. *cebpα*, *cebpγ*, *atf2* and *nf-κB*), signal mediators (e.g. *nfkbiz*, *tnfaip3* and *traf6*), and effector molecules (e.g. complement factors and the sea urchin-specific immune response gene family *185/333*). These data indicate that several cytokines are also transcriptionally regulated in this response, including macrophage inhibitory factors (*mif7*) (*Ch Ho et al., 2016*), TNF superfamily members, and, notably, homologs of IL17 (*Figure 1c*).

To characterize the events that initiate the larval immune response, early changes in gene expression were analyzed by comparing transcript prevalence in larvae exposed to bacteria for 6 hr relative vs. unexposed controls (*Figure 1d*). From this analysis, a small group of *IL17* genes emerge as the most upregulated genes in the genome (*Figure 1d*). Notably, these transcripts are completely absent from transcriptomes assembled from unchallenged (presumably immunoquiescent) larvae (*Tu et al., 2012*). The acute upregulation of these genes suggests that this group of IL17 genes may play a role in initiating the larval response to perturbation of lumenal bacteria. As a foundation for functional study of these cytokines, we next characterized the purple sea urchin IL17 complement from a genomic perspective.

## IL17 homologs encoded in the purple sea urchin genome

Our surveys of the original *S. purpuratus* genome assembly (v2.1) identified 30 IL17-like factors (*Hibino et al., 2006*). However, because many of these homologs were distantly related to each other and also IL17 sequences in other species, we reanalyzed the current genome assembly (v4.2; www.echinobase.org). Using these sequences as queries in BLAST searches, and HMMER analyses to identify IL17 domains (PF06083) in the translated genome sequence, 34 IL17 homologs were identified. Of these, 22 correspond to previously annotated gene models (gene model numbers and coordinates are shown in *Supplementary file 2*). In addition to BLAST (which requires primary sequence similarity) and HMMER (which can be complicated by intron sequences), we scanned uncharacterized but transcriptionally active regions of the genome to identify divergent IL17 factors. RNA-Seq reads were analyzed as they mapped to the genome without consideration of the established gene or transcript models. Genomic regions that exhibited changing expression levels (e.g. were expressed in infected larvae but not in uninfected controls) and lacked any previously described genes were selected. Candidate regions were translated and searched for domains common to immune proteins. One of these expressed, unannotated regions contained a partial IL17 domain. Using the transcriptome data to guide the prediction of coding sequence, a second nearby exon was identified and experimentally confirmed using RT-PCR (*Figure 1e*). The spliced sequence (which is the single member of the *SpIL17-4* subfamily) is divergent relative to the other sea urchin IL17 genes and was not identified using BLAST searches.

The *S. purpuratus* genome (v4.2) thus contains 35 homologs of *IL17* (hereafter referred to as *SpIL17*). Five of these sequences appear to be either pseudogenes that contain premature stop codons or frame shifts (these were verified by reference to raw unassembled trace sequence), or are truncated within the genome assembly due to sequence ambiguity. Phylogenetic analysis of the 30 remaining sequences indicates that this family of genes is comprised of 10 subgroups (designated *SpIL17-1-10*; *Figure 2*). As a complementary analysis to define the echinoderm *IL17* subfamilies and to provide phylogenetic context for the *SpIL17* sequences, we identified *IL17* homologs in five additional echinoderm species that represent a range of taxonomic distances (divergence times of 5–480 million years ago (*Thompson et al., 2015*; *Pisani et al., 2012*; *Biermann et al., 2003*; *Smith et al., 2006*); *Table 1*, *Supplementary file 2*). The three closely related strongylocentrotid species (*S. purpuratus*, *S. fragilis* [formerly *Allocentrotus fragilis* (*Kober and Bernardi, 2013*)], and *Mesocentrotus franciscanus* [formerly *Strongylocentrotus franciscanus* (*Kober and Bernardi, 2013*)]) are estimated to have similar complements of between 30 and 47 IL17 homologs. In contrast, two other sea urchins (the euechinoid *Lytechinus variegatus* and the cidaroid *Eucidaris tribuloides*) and the asteroid *Patiria miniata* each have fewer IL17 genes (7-23; *Table 1*). Phylogenetic analysis of the echinoderm IL17 genes indicates that representatives of the 10 subfamilies are present within each of the euechinoids (*Figure 2—figure supplement 1*). The more distant *E. tribuloides* genome also contains orthologs of most of the *SpIL17* subfamilies (with the exception of groups 3, 9, and 10). The conservation of these families over 260 million years may reflect conserved roles within the immune response. Notably, the *P. miniata* IL17 sequences did not cluster with any of the echinoid sequences, with the

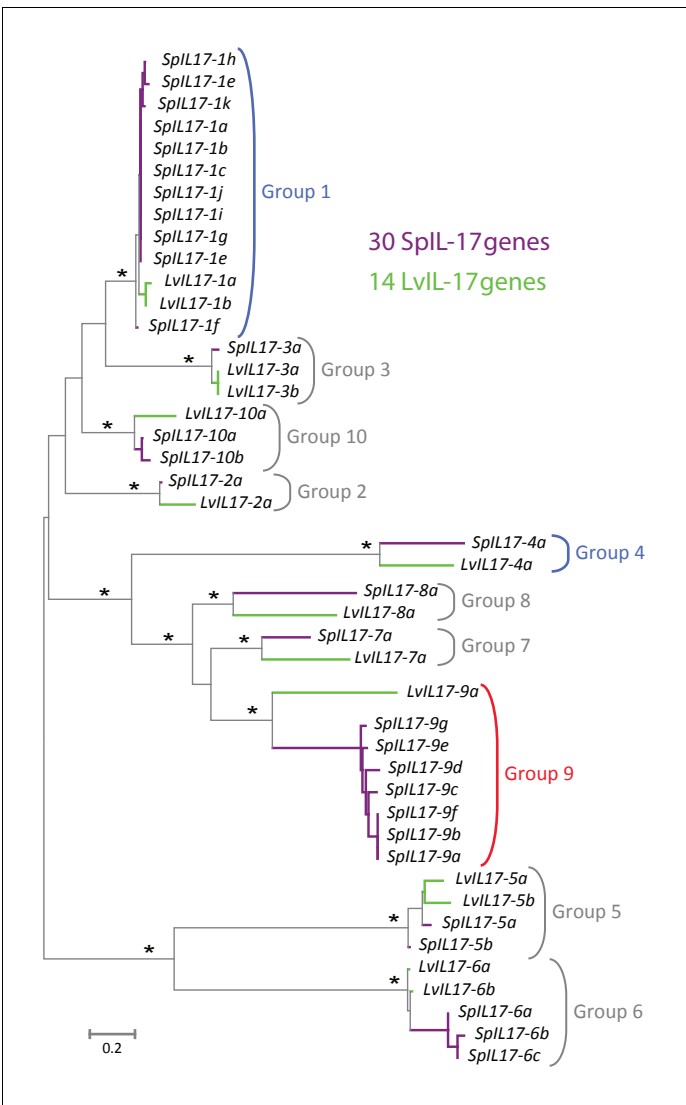

**Figure 2.** Phylogenetic analysis of the sea urchin IL17 sequences defines 10 subfamilies. Predicted amino acid sequences of the 30 IL17 proteins from *S. purpuratus* (purple lines) and 14 IL17 sequences from *L. variegatus* (green lines) were aligned and used in a phylogenetic analysis. The neighbor-joining tree is shown to scale and was constructed in MEGA6.0 using Poisson corrected evolutionary distances, a gamma distribution model for rate variation among sites and complete deletion of alignment positions containing gaps (*Tamura et al., 2013*). Asterisks indicate branches with bootstrap values greater than 75 and based on 500 replicates. Each of the 10 major clades was recovered in phylogenetic trees constructed using maximum likelihood and maximum parsimony methods as well as variable model parameters using the neighbor-joining method (data not shown). Groups are indicated by brackets. Subfamilies expressed in the larval immune response are shown in blue; the family that is expressed in adult immune cells is indicated in red.

The following figure supplement is available for figure 2:

**Figure supplement 1.** Phylogenetic analysis of the echinoderm *IL17* sequences.

exception of *PmIL17-1*, which is related to the *EtIL17-N4* family. The short, highly divergent *IL17* sequences preclude robust phylogenetic analysis beyond the echinoderm lineage. Consequently, orthology cannot be confidently assessed among specific sea urchin IL17 sequences and those from vertebrates.

**Table 1.** Numbers of IL17 genes by subfamily in echinoderm species.

| | Echinodermata | | | | | |
| | Echinoidea | | | | | |
| | Euechinoidea | | | | | |
| | Strongylocentrotidae | | | Toxopneustidae | Cidaroidea | Asteroidea |
| Subfamily | S. purpuratus | S. fragilis* (5–7 myr)[†] | M. franciscanus*[a] (20 myr)[†] | L. variegatus (50 myr)[†] | E. tribuloides (268 myr)[†] | P. miniata (480 myr)[†] |
|---|---|---|---|---|---|---|
| 1 | 11 | 8.6 | 10.0 | 2 | 6 | 0 |
| 2 | 1 | 0.5 | 0.4 | 1 | 1 | 0 |
| 3 | 1 | 0.5 | 0.9 | 2 | 0 | 0 |
| 4 | 1 | 1.9 | 0.4 | 1 | 3 | 0 |
| 5 | 2 | 3.3 | 3.5 | 2 | 2 | 0 |
| 6 | 3 | 4.4 | 4.4 | 2 | 3 | 0 |
| 7 | 1 | 2.9 | 7.8 | 1 | 2 | 0 |
| 8 | 1 | 1.9 | 6.5 | 1 | 2 | 0 |
| 9 | 7 | 13.8 | 7.1 | 1 | 0 | 0 |
| 10 | 2 | 1.0 | 5.2 | 1 | 0 | 0 |
| Other | - | - | - | - | 4 | 12[‡] |
| Total | 30 | 38.6 | 47.0 | 15 | 22 | 12 |

*Estimates are based on the number of best reciprocal blast hits using the SpIL17 sequences against the unassembled genomic trace sequences (**Buckley and Rast, 2012**).

[†]Estimated divergence times shown in million years from *S. purpuratus* (**Thompson et al., 2015**; **Pisani et al., 2012**; **Biermann et al., 2003**; **Smith et al., 2006**).

[‡]See **Figure 2—figure supplement 1** for the phylogenetic analysis of these genes.

Within the purple sea urchin genome assembly, the *SpIL17* genes are located on nine scaffolds (**Figure 3—figure supplement 1**). Each gene is encoded by one to three exons, and all but five are clustered in tandem arrays of two or more genes. The genomic organization of the *IL17* genes is largely conserved between *S. purpuratus* and *L. variegatus* (**Figure 3—figure supplement 1**). To confirm the transcript sequences of the *SpIL17-1* and −4 genes, 5′ RACE PCR was carried out on cDNA generated from larvae 6 hr after infection with *V.d.* Sequences were additionally verified (including the 3′ untranslated regions) using the RNA-Seq data. The *SpIL17-1* genes have two exons, the first of which encodes a methionine and a single glutamate; the IL17 domain is encoded in the second exon (**Figure 3a**). *SpIL17-4a* has two transcripts that initiate at exons with distinct transcription start sites (TSS) but share the second and third exons. These alternative first exons result in different N-terminal sequences that modify the predicted secretion signal peptide (*SpIL17-4a* encodes five amino acids, whereas the alternative *SpIL17-4a′* first exon encodes only a methionine; **Figure 3b**), although the cleavage site is not affected. The functional consequences of this difference are unknown.

The SpIL17 amino acid sequences contain eight conserved cysteine residues (**Figure 3c**). In vertebrates, four of these (highlighted in dark gray in **Figure 3c**) form disulfide bonds at the core of stereotypical cysteine-knot structures (**Hymowitz et al., 2001**). The two C-terminal conserved cysteine residues are absent from subfamilies 3 and 9, which may indicate that these predicted sequences are truncated or that different cysteines are used instead. The vertebrate IL17 sequences also encode two conserved serine residues (**Hymowitz et al., 2001**) that are replaced by shared cysteines in the sea urchin sequences (light gray; **Figure 3c**). Within this framework of conserved amino acids, the SpIL17 sequences exhibit varying levels of diversity within and among the subfamilies. The largest subfamily, *SpIL17-1*, consists of 11 very closely related genes (94.9% average amino acid identity; **Figure 3—figure supplement 2a**). Among subfamilies, these proteins exhibit much higher diversity (an average of 33.3% amino acid identity). For comparison, the six human IL17 proteins share an average 35% amino acid identity (**Figure 3—figure supplement 2a**). Much of this diversity

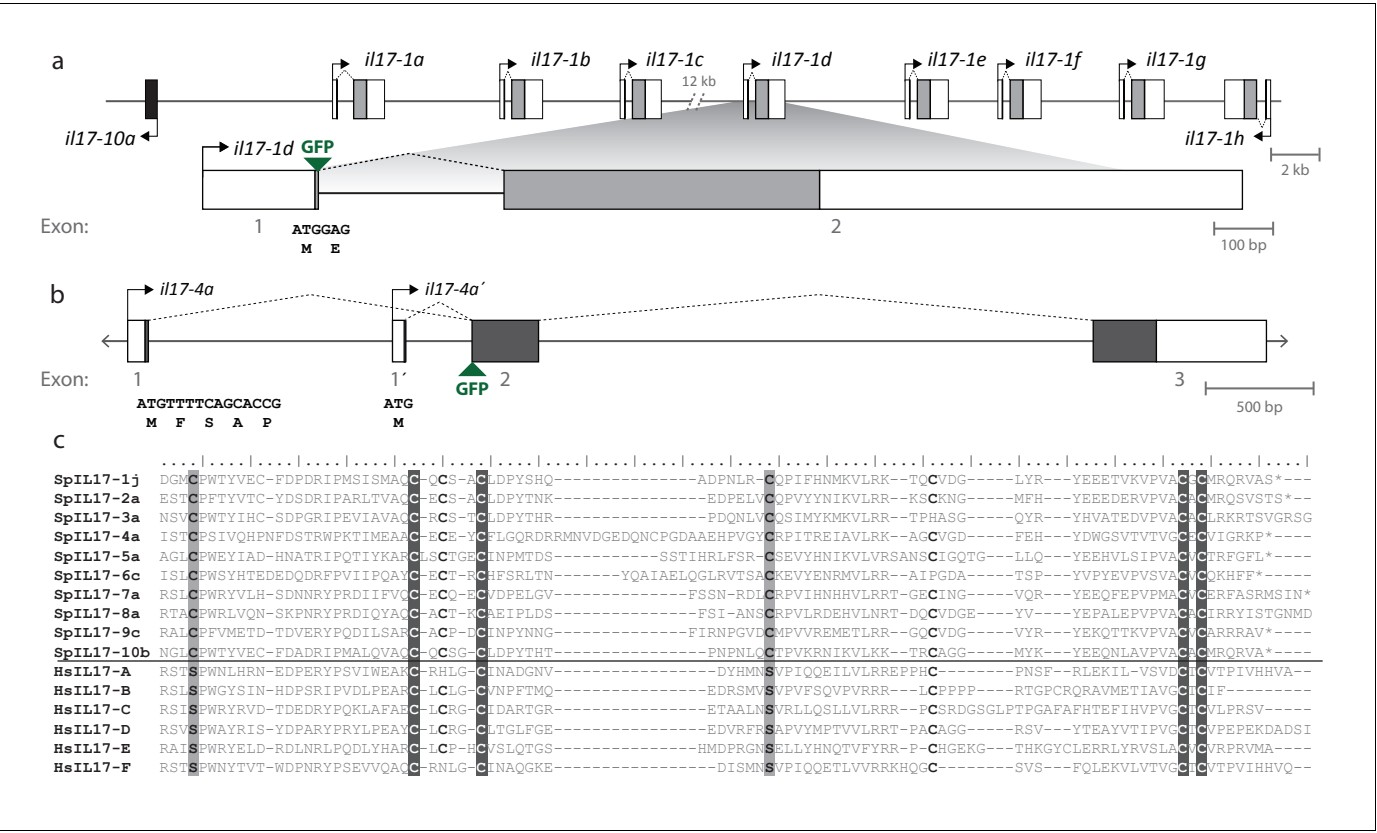

**Figure 3.** Gene structure and diversity of the *SpIL17-1* and *-4* genes. (a,b) Coding sequence is shown in the colored boxes; non-coding sequence is in white boxes. Untranslated regions have been verified using RACE PCR and through analysis of the RNA-Seq data. The genomic structure of all the *SpIL17* genes is shown in *Figure 3—figure supplement 1*. (a) The *SpIL17-1* genes are arrayed in a tandem cluster. The eight *SpIL17-1* genes (light gray) and the adjacent *SpIL17-10a* gene (black) are located on a single scaffold in a 59.8 kb region (Scaffold1147; Genbank KN912785). The *SpIL17-1* genes are encoded in two exons, the first of which includes the methionine and a single amino acid. The entire region is located on BAC clone R3-17F18, which was used to construct a GFP reporter for gene *spIL17-1d*. The position of the GFP in this reporter construct is indicated. (b) The *spIL17-4a* gene encodes two transcripts that initiate from distinct TSS. The nucleotide and translated amino acid sequences are shown for each of the initial exons. (c) The sea urchin and human IL17 proteins share key cysteine residues. The amino acid sequences of the IL17 domains of a member of each of the SpIL17 subfamilies as well as the six human IL17 factors are shown. The conserved cysteine residues implicated in forming the cysteine knot are highlighted in dark gray with white text. Positions in which the SpIL17 proteins have a cysteine that corresponds to a conserved serine in vertebrates are shaded light gray. Additional conserved cysteine residues are indicated in bold.

The following figure supplements are available for figure 3:

**Figure supplement 1.** Genomic organization of the *S. purpuratus* and *L. variegatus IL17* genes.

**Figure supplement 2.** The SpIL17 sequences within subfamilies are highly conserved.

**Figure supplement 3.** Diversity of the IL17 proteins.

is concentrated within the N-terminus of the SpIL17 sequences, consistent with IL17 cytokine family diversity in other groups *Figure 3—figure supplement 2b* (*Pappu et al., 2010*).

## Two SpIL17 subfamilies are strongly activated in response to *Vibrio diazotrophicus* exposure

The RNA-Seq screens of larvae exposed to *V.d* for 0, 6, 12, and 24 hr reveal that genes within two *SpIL17* subfamilies (SpIL17-1 and −4) are sharply upregulated in response to microbial perturbation in the gut lumen (*Figure 1d*). To more thoroughly characterize expression of the *SpIL17-1* and −4 genes, immune challenged larvae were sampled at higher resolution and gene expression was

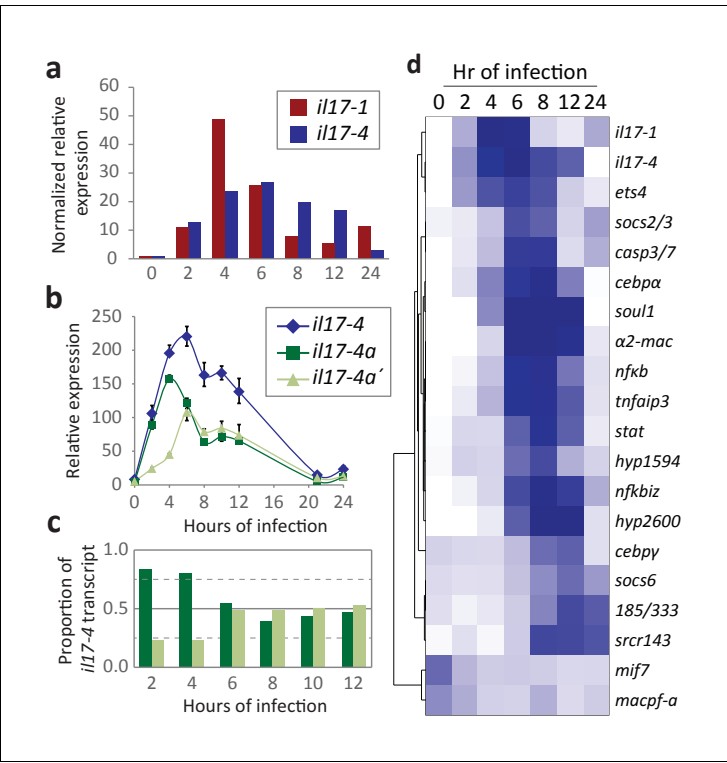

**Figure 4.** Expression of the SpIL17 factors in response to bacterial infection. (**a**). Genes within two *SpIL17* subfamilies are quickly upregulated in response to bacteria. Expression of the *SpIL17-1* (red bars) and *SpIL17-4* (blue bars) genes was measured by RT-qPCR. Relative expression values are normalized to the level of expression in uninfected larvae (0 hr). Non-normalized data with error bars are shown in *Figure 4—figure supplement 1*. Oligonucleotides used in the RT-qPCR reaction anneal to all the *SpIL17-1* genes and both of the *SpIL17-4* transcripts (*Supplementary file 1*). (**b,c**). The two *SpIL17-4a* transcripts are both expressed during infection. Transcript levels of the *SpIL17-4a* (dark green) and −*4a′* (light green) transcripts were measured are shown as expression relative to 18S transcripts (**b**) and as the proportion of total *SpIL17-4* (blue) as measured with primers located in the shared exons 3 and 4 (**c**; see gene structure in *Figure 2b*). Only time points with significant *SpIL17-4* transcript levels are shown in (**c**). (**d**) Activation of the *SpIL17* genes precedes transcriptional changes in many other genes. Transcript prevalence was measured using RT-qPCR for genes that are either known to be involved in immune response in either sea urchins or other organisms. Expression values are log transformed and centered on the mean values for each gene.

The following figure supplement is available for figure 4:

**Figure supplement 1.** Many genes are transcriptionally regulated in larvae responding to microbial perturbation of the gut.

quantified with RT-qPCR (*Figure 4a*). In these samples, the *SpIL17-1* genes are strongly upregulated within 2 hr of bacterial exposure, peak at 4 hr (49-fold higher than at 0 hr) and are downregulated by 8 hr, although expression remains higher than pre-exposure levels (*Figure 4a*). The 11 *SpIL17-1* genes are 87.1–99.8% identical at the nucleotide level. This high similarity, as well as the high level of heterozygosity within the sea urchin population precludes determining expression levels for specific genes using either RNA-Seq data or qPCR. We have isolated and sequenced *SpIL17-1* transcripts from infected larvae using PCR and find that multiple genes are transcribed. Additionally, analysis of single-nucleotide polymorphisms within the RNA-Seq data indicates that most of the *SpIL17-1* genes are biallelically expressed.

The single group four gene, *SpIL17-4a*, is also upregulated in response to bacteria (*Figure 4a–c*). Like the *SpIL17-1* genes, *SpIL17-4a* expression is undetectable in unexposed larvae, is activated by 2

hr of *V.d.* exposure, although its expression peaks slightly later (by 6 hr of exposure). In contrast to the *SpIL17-1* genes, which are consistently downregulated by 8 hr, *SpIL17-4a* expression is downregulated more slowly (*Figure 4a*). While there is some variation in the timing of the downregulation of *IL-17-4a,* these general expression profiles have been reproduced in these and other independent challenge experiments carried out with larvae generated from different mate pairs (e.g. RNAseq in *Figure 1c* and QPCR in *Figure 4a*) and are consistent with qualitative findings from independent *in situ* hybridization time course experiments (*Figure 5a,b*) as well as with from quantitative measurement of GFP reporter transgene expression levels (*Figure 5—figure supplement 1b*). The relative expression levels of the two *SpIL17-4a* transcripts were measured using primers that anneal to sequence in the unique first exons for each transcript and the common second exon (see gene structure in *Figure 2b*) and compared to transcript levels measured using primers located in the shared exons (exons 2 and 3; *Figure 4b,c*). Results indicate that *SpIL17-4a* is upregulated prior to *SpIL17-4a´* (at 2 and 4 hr, *SpIL17-4a* transcripts comprise 84% and 80% of the total; *Figure 4b,c*). From 6–12 hr, however, expression levels are comparable for both isoforms.

To put *SpIL17* expression in the context of other immune factors, we generated expression profiles for additional genes that are known to be important in animal immunity (e.g. the echinoid-specific acute immune effector family *185/333* (*Smith, 2012*; *Figure 4d* and *Figure 4—figure supplement 1*). Analysis of these data reveals that the activation of the *SpIL17-1* and −4 genes is one of the first transcriptional events in the larval immune response. Many changes in expression levels are evident by 6 hr of exposure or later (e.g. *tnfaip3, nfkbiz,* and *cebpα*). Of the 23 genes assayed, a similarly early activation was evident only for *ets4* (SPU_008528, a homolog of the human Prostate-derived Ets transcription factor; PDEF [*Rizzo et al., 2006*]). The early and rapid activation of these cytokines suggests that they may be involved in the initiation of the immune response. Notably, although *de novo* assembly of the RNA-Seq reads recovered spliced transcripts from the *SpIL17-2, −5,–6,* and −9 families, RT-qPCR analysis indicates that these genes are expressed at very low levels and expression is not affected by immune challenge. No expression of genes within the other *SpIL17* subfamilies was evident during the larval immune response. Furthermore, the *SpIL17-1* and −4 transcripts were not present in unchallenged larvae. IL17 domains are also absent from the extensive sea urchin transcriptome databases, which are generated from immunoquiescent tissues and animals. Expression of *SpIL17* genes is therefore tightly regulated and restricted to specific immune challenge conditions.

## SpIL17 expression is restricted to the larval gut epithelium during the immune response to *V. diazotrophicus*

To localize the expression of the *SpIL17* genes within the larval immune response, whole mount *in situ* hybridization (WMISH) was used (*Figure 5a,b*). Data from these analyses are consistent with the temporal kinetics during larval infection described above. No expression of either the *SpIL17-1* or −4 genes was evident in WMISH with uninfected larvae (*Figure 5a1,b1*). *SpIL17-1* expression was observed in the midgut and hindgut of larvae collected at 6 hr of infection (70% of larvae; *Figure 5a,c*). These data suggest that individual larva express the *SpIL17-1* genes for a very short period of time, and the overall increase in expression at 6 hr is the average rate of expression over thousands of larvae. Similarly, expression of *SpIL17-4* is primarily restricted to the mid- and hindguts of larvae exposed to *V.d.* for 6 hr, although some expression is evident at 12 hr (*Figure 5b2, b3*). Fluorescent in situ hybridization using probes for both *SpIL17-1* and −4 indicate that the two IL17 factors are largely co-expressed within the gut epithelium, however, the two transcripts do not always completely overlap (*Figure 5d*).

Many of the finer details of the larval morphology are lost during the fixation process for WMISH. Thus, to localize expression of the *SpIL17* genes *in vivo*, we generated BAC-based fluorescent reporter constructs to recapitulate endogenous expression (*Figure 5e-i*). BAC R3-17F18 spans a 140 kb genomic region that encompasses the eight *SpIL17-1* genes and *SpIL17-10a* on Scaffold1147 (*Figure 3—figure supplement 2*). Using homologous recombination, the GFP coding sequence was inserted into the coding sequence in exon 1 of the *SpIL17-1d* gene (shown in *Figure 2a*). Linearized BACs were injected into fertilized eggs, which were cultured to larval stage (10 dpf) and infected with *V.d.* No GFP expression was observed in larvae prior to infection (*Figure 5e*). By 16 hr of infection, however, fluorescent signal was evident within a few cells within the mid- and hindgut epithelium (*Figure 5f,g*). The lag in visualizing GFP compared to endogenous *SpIL17-1* transcription is

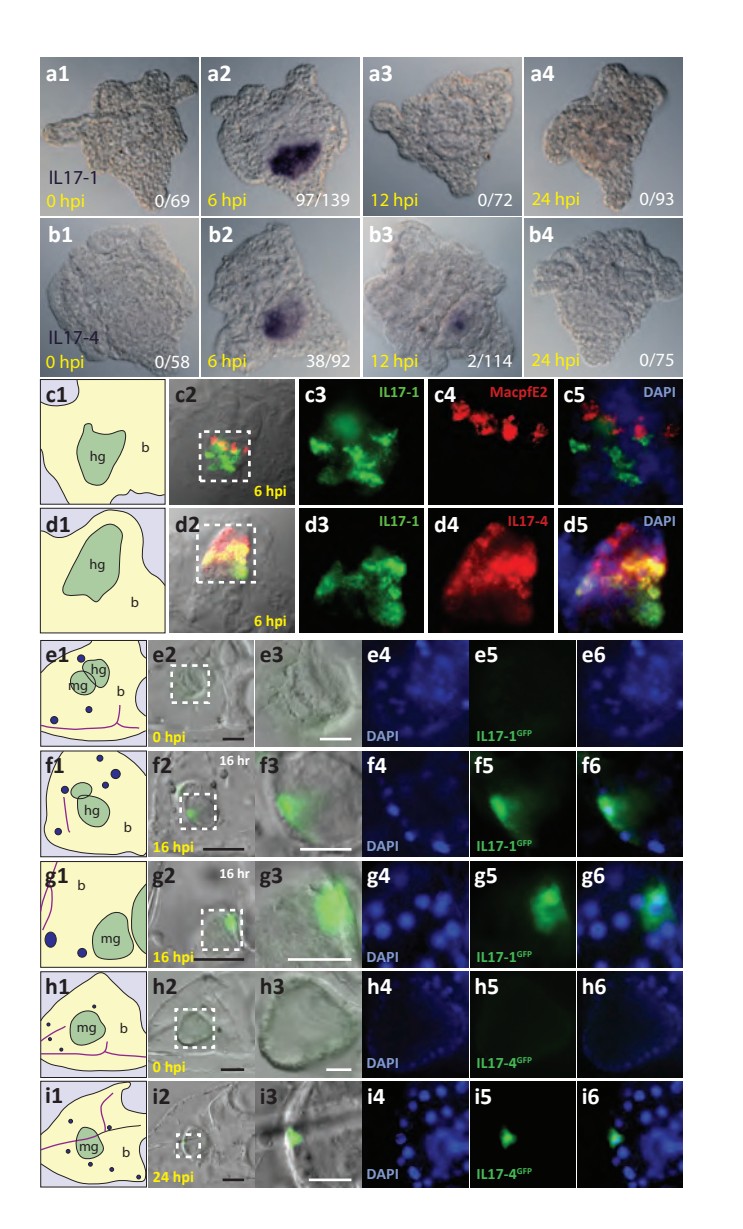

**Figure 5.** The *SpIL17-1* and −4 genes are expressed in gut epithelial cells in response to bacterial challenge. Expression of the *SpIL17-1* (a, c, e–g) and −4 (b, d, h, i) genes was assessed using WMISH (a–d) and BAC-based GFP reporter constructs (e–i). White numbers shown in (a, b) indicate the number of positive larvae out of the total examined. Hours post-infection (hpi) with *V.d.* are indicated in yellow. Larval morphology is shown in c1 – i1 (b, blastocoel, yellow; hg, hindgut; mg, midgut; gut, green; skeleton, purple; blastocoelar immune cells, blue). White dashed lines shown in c2 – i2 indicate the location of the insets. Black scale bars indicate 50 µM; white bars indicate 20 µM.

The following figure supplement is available for figure 5:

**Figure supplement 1.** Transgene reporter constructs recapitulate endogenous *SpIL17-1* expression.

---

consistent, at least in part, with the time required to accumulate and fold the GFP protein. RT-qPCR quantitation of GFP transcript levels normalized to incorporated transgene copy number confirms that the kinetics of GFP expression are similar to that of endogenous *SpIL17-1* expression, with a

sharp increase in expression by 8 hr that is attenuated at 24 hr of infection (*Figure 5—figure supplement 1*).

Similarly, a BAC reporter construct was generated for the *SpIL17-4* transcripts. For this gene, the GFP coding sequence was inserted into the second exon, which is common to both transcripts (see *Figure 2b*) and analyzed as above. This reporter construct also recapitulates endogenous *SpIL17-4* expression. No fluorescence was observed in uninfected larvae, but by 24 hr of infection, GFP was evident in midgut epithelial cells (*Figure 4h,i*), which is delayed relative to *SpIL17-1* as is the endogenous *SpIL17-4*.

Together, these data confirm that the *SpIL17-1* and −4 genes are expressed exclusively in the epithelial cells of the larval mid- and hindgut, and that expression of these genes is tightly regulated and dependent upon bacterial challenge.

## A third SpIL17 subfamily is upregulated in adult immune cells

To investigate the role of the *SpIL17* gene family in the adult sea urchin immune response, we analyzed RNA-Seq data collected from adult immune cells (phagocytic coelomocytes) and gut tissues isolated following immune challenge (*Buckley and Rast, 2012*). The tissues used in this experiment were collected from a single animal that was injected intracoelomically with bacteria isolated from the gut lumen of another individual. This complex challenge mimics a gut perforation and generally increases the expression of many genes involved in immunity. Analysis of the RNA-Seq data indicated that while *SpIL17* expression was not detected in the gut tissue, genes within a third subfamily, *SpIL17-9*, were expressed at low levels in coelomocytes at 12 hr after challenge.

To determine whether this low level of expression reflected a larger transcriptional response to immune challenge, adult animals were challenged by intracoelomic injection of either *V.d.* or sham controls (seawater injection). Coelomocytes were collected at six time points over the course of 24 hr and used for gene expression analysis (*Figure 6*). Expression of the sea urchin immune response genes *185/333* were used to assess immune activation. This echinoid-specific family of diverse defense genes is strongly upregulated in response to several types of immune challenge (*Ghosh et al., 2010*). RT-qPCR analysis indicates that the *SpIL17-9* genes are strongly upregulated within 3 hr of infection in both animals injected with bacteria (red bars; *Figure 6a*). Expression peaks at 6 hr, and then returns to lower levels. This timing precedes expression of the *185/333* genes, which are slightly upregulated at 3 hr of infection, but exhibit high levels of expression by 9 hr (*Figure 6b*). In the animal that received the sham seawater injection, expression of the *SpIL17-9* genes also increased, but often more slowly (expression peaked at 12 hr; *Figure 6a*). This is consistent with a later activation of the *185/333* genes in this animal (24 hr; *Figure 6b*). A slower, more attenuated response is typical in sham-injected animals (*Rast et al., 2000*). Additionally, in the experiments described here, the repeated needle sticks during the time course sampling may also elicit an acute injury response even in the absence of injected bacteria. Nonetheless, these experiments demonstrate that adult phagocytic cells can be induced to express the *SpIL17-9* subclass and that it is silent in undisturbed animals.

We found no evidence of expression of *SpIL17-9* genes in larvae responding to *V.d.* exposure, and *SpIL17-1* or −4 expression was never evident in the adult coelomocytes. Together, these data indicate that genes within at least three of the *SpIL17* subfamilies are expressed in the course of the sea urchin immune response and that the subfamilies are deployed in distinct tissues during the different life stages, although it remains to be seen if different modes of challenge will lead to other expression patterns.

## Interfering with IL17 signaling affects downstream gene expression

To characterize the role of IL17 signaling within the immune response, we searched the *S. purpuratus* genome for homologs of the IL17 receptor (IL17R). Vertebrate IL17 receptors are characterized by a conserved cytoplasmic Sef/IL17 receptor (SEFIR) domain (PF08357) that is structurally similar to the Toll/IL-1 receptor (TIR) domain (*Novatchkova et al., 2003*) but is uniquely associated with IL17 signaling. The SEFIR domain mediates intracellular signaling through interactions with the adaptor molecule Act1, which also contains a SEFIR domain (*Qian et al., 2007*). Mammalian IL17RA is also characterized by a TIR-like loop (TILL) domain, and a loosely defined C/EBP$\beta$ activation domain

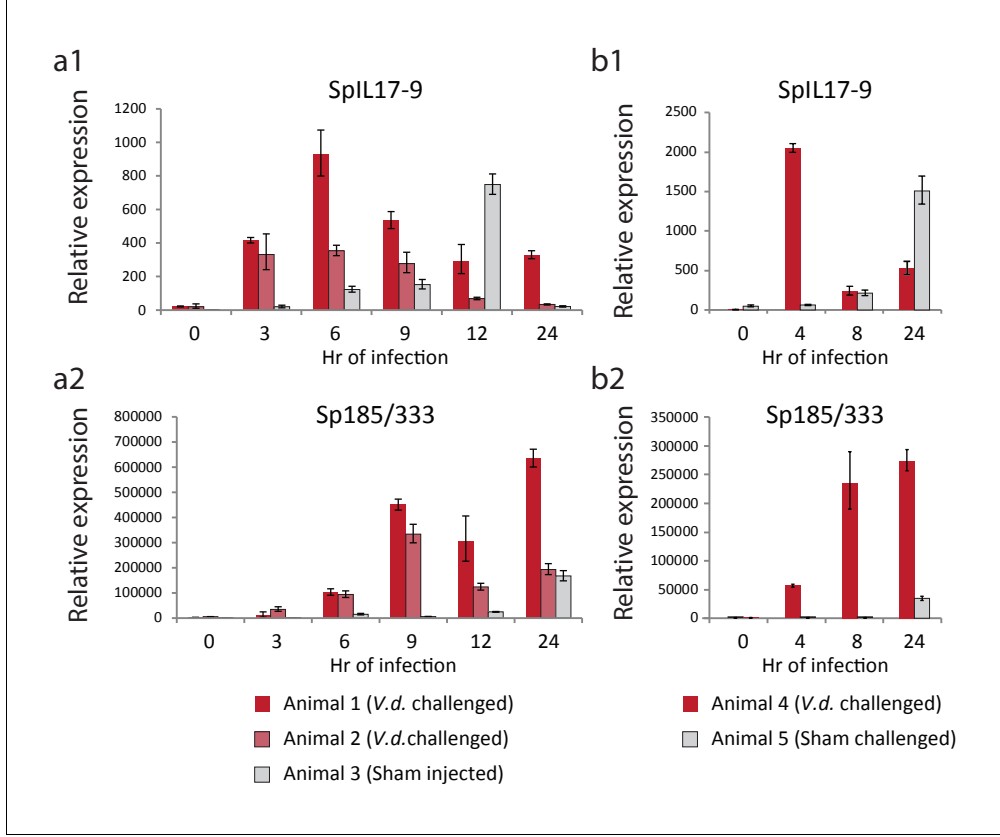

**Figure 6.** The *SpIL17-9* genes are expressed in adult coelomocytes. Data from two independent experiments are shown (**a,b**). qPCR was used to measure transcript prevalence in coelomocytes collected from adult animals that were either injected with live *V.d.* (animals 1, 2, and 4) or sham injection controls (ASW only; animals 3 and 5). The treatment for each animal is indicated below the graphs. Expression levels for the *SpIL17-9* genes (a1, b1) increase strongly by 3 hr of exposure to bacteria, and more slowly in response to injury. Expression of the effector genes *185/333* (a2, b2) serves as a marker of immune activation (*Ghosh et al., 2010*).

(CBAD) that are C-terminal to the SEFIR domain and are required for downstream signaling (*Maitra et al., 2007*).

We find that two sea urchin genes encode SEFIR domains (SpIL17R1 and SpIL17R2; *Figure 7*; *Figure 7—figure supplement 1a,b*). Each of these genes encodes a signal sequence, a long putative extracellular region (535 or 662 amino acids), a transmembrane region, and a cytoplasmic SEFIR domain. This structure is consistent with IL17 receptors in other lineages. Additionally, phylogenetic analysis of the sea urchin SEFIR domains supports homology with IL17 receptors in other species (*Figure 7—figure supplement 1*). We have confirmed these sequences by amplifying the receptors using PCR and sequencing. A TIR-like loop (TILL) sequence is present in SpIL17R1 directly C-terminal to the SEFIR domain but is absent in SpIL17R2 (*Figure 7*). SpIL17R1 is encoded in 17 exons (*Figure 7—figure supplement 3a*). Exon 16, which encodes the sequence between the transmembrane region and the SEIFR domain is alternatively spliced and is absent from some transcripts. Similarly, SpIL17R2 is expressed in 16 exons; the last exon encodes the SEFIR domain (*Figure 7—figure supplement 3a*).

We characterized the temporal expression of the SpIL17 receptor genes in developing embryos and larvae as well as adult tissues. qPCR indicates that *SpIL17R1* expression is not evident at 12 hpf, but increases slowly to prism larval stage (84 hpf). *SpIL17R2* is upregulated at 24 hpf, peaks at 48 hpf, and then returns to low levels into the larval stage (*Figure 7—figure supplement 3c*). In larvae, *spIL17R1* is gradually downregulated during infection, whereas SpIL17R2 expression is mostly constant but exhibits a small peak at 2 hr of infection and then again at 24 hr (*Figure 7—figure*

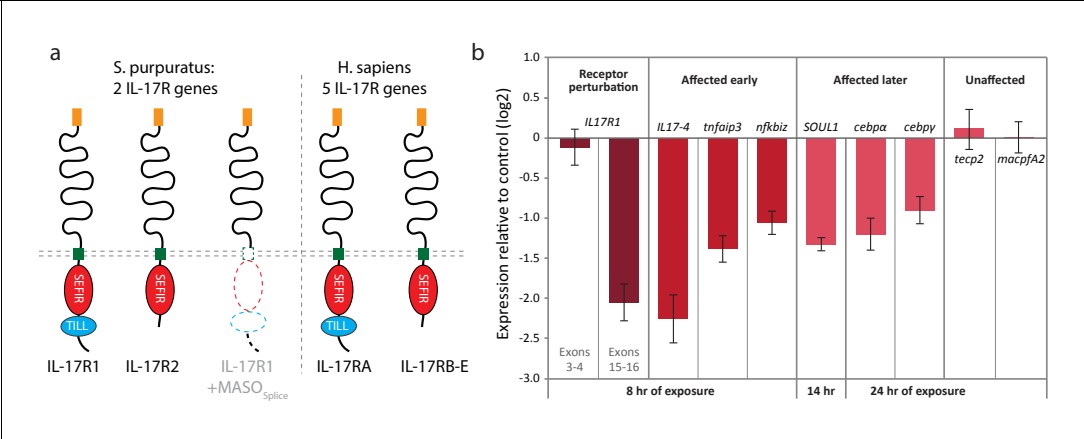

**Figure 7.** Two IL17 receptors mediate IL17 signaling in the sea urchin. (a) The sea urchin IL17 receptor sequences have similar domain architectures as those in vertebrates. The sea urchin receptors have conserved SEFIR domains (red). SpIL17R1 also has a TILL domain (blue). The structure of the protein encoded by the *SpIL17R1* transcript in the presence of the splice-blocking MASO (MASO$_{Splice}$) is also shown. This MASO interferes with splicing by binding to the donor splice site in exon 15 (see *Figure 7—figure supplement 2*). Consequently, a cryptic donor splice site in exon 14 is used, which introduces a frameshift and premature stop codon. The resulting truncated protein does not contain a transmembrane or SEFIR domain (indicated by white shapes). (b). Interfering with IL17 signaling affects the expression of downstream genes during immune challenge. Fertilized eggs were injected with the IL17R1 MASO$_{Splice}$ and grown to 10 dpf. Larvae were infected with *V.d.* and collected for RT-qPCR analysis. Complete data are shown in *Figure 7—figure supplement 3*.

The following figure supplements are available for figure 7:

**Figure supplement 1.** Phylogeny of SEFIR domain-containing proteins.

**Figure supplement 2.** The splice-blocking SpIL17R1 MASO$_S$ yields a transcript with a frameshift and premature stop codon.

**Figure supplement 3.** The *S. purpuratus* genome encodes two IL17 receptors.

**Figure supplement 4.** Effects of IL17R1 perturbation on downstream gene expression.

*supplement 3d,f*). To assess transcript prevalence in adult tissues, we analyzed publicly available RNA-Seq data (generated from adult coelomocytes, axial organ, gut, radial nerve, ovary and testes [*Tu et al., 2012*]). Results indicate that low expression of *spIL17-R1* is evident in adult gut and testes. The *spIL17r2* transcript was not clearly present in any of the adult tissues assayed.

Despite extensive searches, we were not able to identify a homolog of Act1 (the SEFIR domain-containing adaptor molecule for the IL17 receptor in vertebrates [*Qian et al., 2007*]) in any of the echinoderm genomes. Other than the two SpIL17 receptors, we found no evidence of additional sequences that encode SEFIR domains in the purple sea urchin genome (including all gene models, open reading frames and transcriptome data). Similarly, the *L. variegatus* genome contains two IL17 receptors and lacks an Act1 homolog (*Figure 7—figure supplement 1*). Given the close relationship between the SEFIR and TIR domains, we analyzed the seven unique TIR domain-containing molecules within the *S. purpuratus* genome (*Hibino et al., 2006*), but none of these exhibit sequence or domain similarity to the Act1 proteins.

To assess the role of IL17 signaling in the sea urchin larval immune response, we perturbed SpIL17R1 using two Morpholino antisense oligonucleotide (MASO) reagents: one that interferes with translation by annealing at the translation start site (MASO$_T$) and a splice-blocking MASO that binds to the donor splice site in exon 15 (MASO$_S$; *Figure 7—figure supplement 3*). Because these two MASOs resulted in similar effects, we concentrated on the splice-blocking MASO, as the efficiency of this treatment could be directly quantified using qPCR. Translation blocking morpholinos generally do not affect message prevalence of their targets in the absence of regulatory feedbacks (e.g. [*Rast et al., 2002*; *Davidson et al., 2002*]). To confirm the effect of this splice-blocking MASO, we

amplified and sequenced the *SpIL17R1* exons 14 through 17 from transcripts expressed in larvae (10 dpf) subjected to MASO$_S$ (*Figure 7—figure supplement 2*). Analysis of these sequences indicates that treatment with the MASO$_S$ produces incorrectly spliced *SpIL17R1* transcripts, in which an alternative donor site within exon 14 is spliced directly to exon 16, resulting in a frame shift and premature stop codon. The subsequently translated protein does not encode a transmembrane domain (*Figure 7a*). To assess the efficacy of this MASO, qPCR was performed on larvae using oligonucleotides that bind to a series of *SpIL17R1* exons (*Figure 7—figure supplement 3g*). The MASO$_S$ specifically affects the transcription of exons 14 and 15, such that amplification between exons 14/15, 15/16 and 15/17 is significantly lower than in control larvae, whereas amplification between exons 3/4 and 16/17 is unaffected. The MASO$_s$ does not exhibit complete penetrance, however, as low levels of exon 15 transcript amplification are evident in perturbed larvae. This may result in a partial effect on the transcription of downstream genes. Perturbation of SpIL17R2 was developmentally lethal and therefore not pursued in this study (data not shown). This phenotype, however, is consistent with previously described developmental functions for IL17-RD (Sef) (*Tsang et al., 2002*; *Ron et al., 2008*),

Perturbation of IL17 signaling during the larval immune challenge does not significantly affect pigment cell migration (*Figure 7—figure supplement 3h*); however, it does affect the message prevalence of several genes that were chosen as candidate response genes from the timing of their upregulation (*Figure 7b*, *Figure 7—figure supplement 4*). Notably, this includes *SpIL17-4*, which is upregulated relative to uninfected larvae, but at 2–5-fold lower levels than in larvae injected with control MASO. This decreased expression, in addition to the fact that *SpIL17-1* expression consistently lags that of peak *SpIL17-4* (*Figure 4d*) suggests that the SpIL17-1 signaling may be involved in upregulating *SpIL17-4*. Additionally, transcript levels of *tnfaip3*, *soul1*, *nfkbiz*, *cebpα*, and *cebpγ* are reduced in larvae exposed to the SpIL17R1 MASOs compared to control MASO (*Figure 7b*, *Figure 7—figure supplement 4*). These data begin to delineate how early expression of SpIL17 factors in response to bacterial challenge mediates the expression of downstream genes in an intact organism.

## Discussion

### Tightly regulated IL17 expression is central to initiating the immune response

We present here a characterization of the sea urchin larval immune response to microbial perturbation in the gut from a transcriptional perspective. RNA-Seq screens of larvae exposed to bacteria reveal that two subfamilies of IL17 act as key components of this immune response. The *SpIL17-1* genes are acutely upregulated during the earliest phases of immune response. This change in gene expression (~90 fold higher at 6 hr of exposure to *V.d.* relative to unexposed controls that have minimal *SpIL17-1* transcript prevalence) is greater than any other gene at any time point. This early expression pattern may point to a role in activating the downstream immune response and communicating the state of the gut lumen to both other cells of the gut epithelium and to the wider organism. In addition to the IL17 cytokine family, we have characterized two orthologs of IL17 receptors within the *S. purpuratus* genome. Perturbation of these receptors using antisense reagents results in reduced expression of immune genes and transcription regulators, including the *SpIL17-4* genes. These results further support the role of *IL17* signaling at the initiation of the sea urchin larval gut-associated immune response. Finally, the purple sea urchin IL17 subfamily structure is conserved throughout a phylogenetic range of representative echinoderm species, suggesting an ancient origin for the sequence diversity of the echinoderm IL17 sequences.

The conservation of the IL-17 subfamilies may indicate functional compartmentalization for these genes within the echinoderm immune response. This is supported by the tightly regulated expression of the *SpIL17* genes both spatially (gut epithelium in the larva and immune cells of the adult) and temporally (expression is strictly dependent on immune challenge or injury). Of the 10 *SpIL17* subfamilies, genes within three families (*SpIL17-1*, −4, and −9) are expressed in the context of either the larval or adult immune systems. It is likely that genes with the remaining subfamilies are expressed under different conditions of immune challenge or stress. This is particularly true for subfamilies *SpIL17-2*, −5, and −6, which can be detected as spliced transcripts from the larval

transcriptome assembly, although RT-qPCR indicates that these are present at very low levels. The *SpIL17* transcripts are virtually absent under immunoquiescent conditions. We have analyzed available transcriptome data from 16 developmental stages (egg through juvenile) and six adult tissues (*Tu et al., 2012*) and find no evidence for expression of any of the *SpIL17* subfamilies. Furthermore, despite the availability of large EST databases (there are >350,000 echinoderm ESTs from several adult tissues and developmental time points), no ESTs have been identified that correspond to the *SpIL17* sequences. This lack of *SpIL17* expression in non-challenged tissues correlates with our observations and underscores the importance of analyzing transcriptional activity under varied conditions of challenge when targeting immune-related genes.

## Sea urchin genomes encode moderately expanded IL17 families

Data presented here indicate that genome sequences from echinoderms (particularly the strongylocentrotids) have an unusually large number of genes encoding IL17 homologs. Mammals typically have six IL17 orthologs and the genomes of teleost fish that have been analyzed contain between four and seven IL17 genes (*Secombes et al., 2011*). The presence of 35 IL17 genes within the genome of the purple sea urchin suggests that this gene family has been expanded within this lineage. Although the driving forces behind this expansion remain unknown, it is consistent with other large immune-related gene families within the purple sea urchin genome that have both pathogen recognition and regulatory functions. The purple sea urchin genome contains expanded families of pattern recognition receptors that are 10-fold larger than their vertebrate counterparts (*Rast et al., 2006*; *Messier-Solek et al., 2010*). High multiplicity is also apparent among gene families that encode immune effector genes (*e.g.*, the *185/333* gene family [*Ghosh et al., 2010*] and the perforin-like Macpf family [*Hibino et al., 2006*]).

Most of the echinoid IL17 subfamilies are conserved at least to the last common cidaroid-euechinoid ancestor (*Figure 2—figure supplement 1*). Although the *L. variegatus* genome has fewer IL-17 genes, homologs representative of each of the subfamilies are retained. This phylogenetic analysis indicates that lineage-specific tandem duplications also contribute to the diversity of echinoderm IL17 family (e.g. the *SpIL17-1* homologs; *Figure 3—figure supplement 1*). Broader phylogenetic comparisons (*e.g.*, comparing the echinoderm and chordate IL17 sequences) are rendered uninformative by the relatively short and divergent sequences. In these analyses, genes tend to cluster within phyla with low confidence.

## Epithelial expression of IL17: an ancient role in gut-associated immunity

In mammals, studies of the IL17 family are largely focused on expression of IL17A and IL17F in lymphocytes. The Th17 and $\gamma\delta$ T cells are major sources of IL17A and IL17F in response to infection (*Littman and Rudensky, 2010*; *Roark et al., 2008*). Other cell types also produce IL17A and IL17F, including ILCs and myeloid cells (*Roark et al., 2008*; *Passos et al., 2010*; *Michel et al., 2007*; *Takatori et al., 2009*; *Zhu et al., 2008*; *Li et al., 2010*; *Hueber et al., 2010*), and specialized gut epithelial cells known as Paneth cells (*Takahashi et al., 2008*). More recent work, however, has shown the importance of epithelial expression of another IL17 ortholog, IL17C, in directing immune responses in the gut. IL17C is produced by the gut epithelium where it acts in an autocrine manner to activate expression of genes involved in the innate immune response, including proinflammatory cytokines and antimicrobial peptides (*Song et al., 2011*; *Ramirez-Carrozzi et al., 2011*). This is observed in dextran sodium sulfate-induced colitis models as well as infection with *Citrobacter rodentium*. Members of the IL17 family have been implicated in inducing neutrophil migration (*Ye et al., 2001*), regulating tight junction formation (*Kinugasa et al., 2000*; *Reynolds et al., 2012*) and stimulating mucin production (*Chen et al., 2003*). IL17C plays a role in maintaining intestinal barrier integrity by regulating the expression of occludin, a tight junction protein in colonic epithelial cells (*Reynolds et al., 2012*). IL17C expression in the gut epithelium has also been linked to autoimmunity (*Chang et al., 2011*) and tumorigenesis (*Song et al., 2014*). This cytokine is thus a primary mediator in mammalian gut-associated immune response.

Data on IL17 function remain limited outside of the jawed vertebrates. In the lamprey, five IL17 homologs are differentially expressed on skin, kidney, intestine and gills, as well as VLRA[+], VLRB[+] and VLRC[+] lymphocytes (*Han et al., 2015*). *Ciona intestinalis* upregulates three homologs of IL17 in the pharynx and in immune cells in response to LPS challenge (*Vizzini et al., 2015*). In the oyster,

which is the only protostome in which the IL17 response has been characterized, the single IL17 homolog is expressed by circulating hemocytes in response to infection (*Roberts et al., 2008*). In the work presented here, we show that in the sea urchin larva, SpIL17 expression is restricted to gut epithelial cells, with no evidence of expression in mesodermally derived immune cells in response to *V.d.* challenge. This function is potentially homologous to the mammalian epithelial expression and suggests that the role of IL17 in modulating mucosal immunity is an ancient and fundamental component of immunity.

It is notable that, although there is a strong expression of *SpIL17-9* in adult coelomic immune cells in response to immune challenge, over the course of many *V.d.* challenge experiments, we have never observed *SpIL17* expression in any larval mesodermal immune cells. This differential expression pattern may reflect the mode of infection with *V. d.* Ongoing work with different isolated bacteria suggests that other larval cells may be capable of expressing *SpIL17-1* and *SpIL17-4* in altered infection conditions. Additionally, genes within the other IL17 subfamilies may also be expressed in the larva in response to differential immune challenge.

### IL17 receptors

The sea urchin genome encodes two IL17 receptor chains. These contain SEFIR domains with structures that mirror the two types of receptors found in mammals. *In situ* hybridization indicates that the gut epithelium is a primary site of *SpIL17R1* expression (*Figure 7—figure supplement 3e*). Analysis of cell-specific transcriptome data from larvae indicates that this receptor is also expressed in gcm[+] pigment cells (*Barsi et al., 2014*). As in other systems, these receptors may be widely expressed at low levels (*Gaffen, 2009*).

We were unable to detect an Act1 homolog in any echinoderm genome, although homologs are readily detectable in hemichordates and invertebrate chordates (*Ryzhakov et al., 2011*). This suggests that the sea urchin IL17 receptors signal through an Act1-independent mechanism. One potential mechanism identified in mammals is the direct activation of STAT5 that occurs in vertebrate IL17RB signaling (*Wu et al., 2015*). The recruitment of STAT5 depends on phosphorylation of specific tyrosine residues within the IL17RB cytoplasmic region. Specifically, STAT5 recruitment is mediated by tyrosine residues 444 and 454. These residues are both conserved in the SpIL17-R1 sequence. There are 13 additional tyrosine residues in the cytoplasmic region of SpIL17R1 and six in SpIL17R2 that may serve similar functions. Alternatively, a novel mechanism may function within the echinoderm SpIL17 system.

### Potential interactions within the larval IL17 system

Several observations are consistent with feedback in the larval gut epithelial system. The IL17R1 is expressed in the gut epithelial cells consistent with the possibility that neighboring cells in the epithelium could communicate using the IL17-1 signal. In addition, the single *IL17-4* gene is consistently activated to peak levels several hours after the IL17-1 genes in immune challenge experiments even when these vary in time of initial expression. Reduction of IL17-4 expression in IL17R1 MASO perturbed embryos supports a causal linkage between IL17-1 activation and later IL17-4 expression. Thus, IL17-4 may in some way modify signaling initiated by IL17-1. These possibilities can be explored in future perturbation experiments.

### Conclusions

These findings reveal that epithelial IL17 signaling is an ancient and central element of the gut associated immune response. By exploiting the experimental strengths of the morphologically simple sea urchin larva, these findings provide a novel perspective on the regulation and downstream consequences of this highly studied immune signaling factor. Further investigations into the activation and interplay of the IL17 subfamilies within the larval immune response will continue to yield valuable insights that can be applied directly to the more complex mammalian systems.

# Materials and methods

## Animals and larval cultures

*S. purpuratus* animals were obtained from the Point Loma Marine Invertebrate Lab (Lakeside, CA). Animals and larval cultures were maintained and exposed to *Vibrio diazotrophicus* as previously described (*Ch Ho et al., 2016*). To challenge adult sea urchins, *V. diazotrophicus* were cultured in LB at 15°C, washed three times with artificial sea water (Instant Ocean; ASW) and resuspended in 0.2 µM filtered ASW. Animals were injected with $10^5$ bacteria/mL coelomic fluid in a total volume of 500 µL. Sham injected animals were injected with an equal volume of 0.2 µM filtered ASW.

## RNA isolation, quantitative PCR (RT-qPCR) and transcriptome sequencing

Coelomocytes were harvested from adult sea urchins by inserting a preloaded syringe (with a 22-gauge needle) into the peristomial membrane and extracting coelomic fluid as in *Buckley and Smith (2007)*. To prevent clotting, the syringe was preloaded with ice cold calcium/magnesium-free seawater (454 mM NaCl, 9.4 mM KCl, 48 mM $MgSO_4$, 6 mM $NaHCO_3$, pH 7.4). Coelomocytes were pelleted and resuspended in Trizol (Invitrogen). Total RNA was isolated with Trizol (Invitrogen). Contaminating genomic DNA was removed using the DNA-free kit (Ambion). First-strand cDNA was synthesized from random hexamers using Superscript III (Invitrogen). Quantitative PCR was carried out as described (*Rast et al., 2002*; *Fugmann et al., 2006*). Measurements were made in triplicate on a ViiA7 real-time PCR machine using SYBR green chemistry (Applied Biosystems) and expression levels were normalized to parallel 18S rRNA measurements made on samples diluted 1:1000. Primer sequences are shown in *Supplementary file 1*.

Whole transcriptome sequence data was generated for larvae exposed to *V. diazotrophicus* for 0, 6, 12, and 24 hr (*Buckley and Rast, 2012*). Data are available at NCBI (BioProject PRJNA380184). Reads were mapped to the *S. purpuratus* genome (v3.1; www.echinobase.org) using Bowtie, version 0.12.7 (*Langmead et al., 2009*) with modified parameters to accommodate both the polymorphic sea urchin genome as well as the large families of highly similar immune genes that are relevant for this analysis (*Buckley and Rast, 2012*). To assess expression in adult tissues, RNA-Seq data was downloaded from NCBI for project PRJNA81157 (*Tu et al., 2012*) (axial organ, SRX173268; coelomocytes, SRX173270; gut, SRX173274; radial nerve, SRX173280; ovary, SRX173277; testes, SRX173283). Gene expression in tissues collected from an immune activated adult was also assessed using RNA-Seq methods (PRJNA381801). Reads were mapped to the *S. purpuratus* genome (v3.1) as above. Expression levels quantification were performed using Cufflinks (*Trapnell et al., 2012*). De novo transcriptome assemblies were done using Trinity (*Haas et al., 2013*).

To identify novel genes involved in the immune response, the Bowtie output files from the analysis of the larval RNA-Seq experiments were analyzed directly. Output files (in SAM format) were sorted by scaffold and position (using the Linux sort function). Based on the average size of exons within the *S. purpuratus* genome (100–115 nt [*Sodergren et al., 2006*]), numbers and orientations of reads that mapped within 200 nt regions were tabulated (using a sliding window with a 100 nt overlap). Genomic positions that included known gene or transcript models were excluded from further analysis. Bins that contained at least 20 reads of which at least 90% were in the same orientation were ranked by expression level. These sequences were translated and searched for immunologically relevant domains using HMMER *Eddy, 1998*.

## Amplification, cloning and sequencing of Sp-IL17 ligands and receptors

Primer sequences that were used for cloning the SpIL17 ligands and receptors are located in *Supplementary file 1*. Complete cDNA sequences were obtained for the SpIL17 ligands and receptors with RACE PCR using the GeneRacer kit (Invitrogen). Amplified sequences were cloned into pCR-TOPO4 (Invitrogen) and sequenced.

## Whole mount *in situ* hybridization (WMISH)

Infected larvae were washed twice with ASW and fixed overnight in 4% paraformaldehyde, 32.5 mM MOPS pH 7, 32.5% ASW, 162.5 mM NaCl (*Minokawa et al., 2004*). Larvae were washed five times in MOPS buffer (100 mM MOPS pH 7; 500 mM NaCl; 0.1% Tween-20), dehydrated and stored in

70% ethanol at −20°C until use. WMISH was performed as described (colorimetric [*Minokawa et al., 2004*; *Ransick et al., 2002*]; fluorescent [*Croce and McClay, 2010*]).

## BAC reporter constructs

Reporter constructs were generated using homologous recombination (*Yu et al., 2000*) for the *SpIL17-1* gene *SpIL17-1e* using BAC clone R3-17F18 (GenBank: AC201380.1) and for the *SpIl17-4a* gene using BAC clone R3-4009B23 (GenBank: AC179066.1). Primer sequences used to design the recombination arms are shown in *Supplementary file 1*. Recombinant BACs were linearized and microinjected into fertilized eggs at 100–200 copies/pL. Injected larvae were cultured to 10 days, infected with *V. diazotrophicus* as described (*Ch Ho et al., 2016*) and imaged to assess fluorescent reporter expression.

## Morpholino antisense oligonucleotides (MASOs)

Fertilized eggs were injected with MASO reagents (Gene Tools) at a final concentration of 200 µM as described (*Solek et al., 2013*). The MASO sequences are as follows: SpIL17R1 translation-blocking, 5′-GTGACGACATGTGAACCATGGACAT-3′; SpIL17R1 splice-blocking, 5′- CCATTGTTCCCAAA-CACCTACCACT-3′; SpIL17R2 translation-blocking 5′-ACACGATTGCGACGGTGGTTAACAT-3′.

## Phylogenetic analysis and bioinformatics

Genome sequences for *S. purpuratus* (v4.2), *L. variegatus* (v2.2) and *E. tribuloides* (v1.0) and unassembled trace sequences from *M. franciscanus* and *A. fragilis* were obtained from Echinobase (www.echinobase.org) (*Cameron et al., 2009*). The genome sequence from *P. miniata* (v1.0; Bioproject PRJNA49323) was obtained from NCBI. IL17 multiplicity in unassembled genome sequences was estimated by identifying traces with similarity to full-length echinoderm IL17 sequences using BLAST. The number of traces was normalized using the estimated coverage of the genome sequence (*S. fragilis*, 2.1×; *M. franciscanus*, 2.3×).

Tools within the EMBOSS suite were used to translate genomic sequence and identify open reading frames (emboss.sourceforge.net). Domain predictions were done using HMMER (*Eddy, 1998*) using the IL17 PFAM domain (PFAM accession number PF06083) and the SEFIR domain (PF08357). Signal peptides were predicted with SignalP3.0 (*Bendtsen et al., 2004*). Alignments were edited using Bioedit (*Eddy, 1998*). Phylogenetic analyses were performed in MEGA, version 6.0 (*Tamura et al., 2013*). Genbank accession numbers for the IL17 sequences used in as BLAST queries and in phylogenetic analysis are as follows: human IL17A, AAR23263.1; human IL17B, CAG33473.1; human IL17C, AAQ88835.1; human IL17D, AAQ89471.1; human IL17E, AAQ89484.1; human IL17F, AAK83350.1; zebrafish IL17c, NP_001018624.1; zebrafish IL17a/f1, NP_001018623.1; zebrafish IL17d, NP_001018625.1; zebrafish IL17a/f2, NP_001018634.1; zebrafish IL17a/f3, NP_001018626.1; oyster IL17, EW779442.1. Protein structure predictions were performed using Phyre2 (*Kelley and Sternberg, 2009*). Protein sequence identities were calculated with Matgat (*Campanella et al., 2003*).

## Acknowledgements

We thank Michele Anderson for a thorough reading of this manuscript and Andrew Cameron of the Center for Computational Regulatory Genomics for providing the BAC clones. This work was supported by grants from the Canadian Institutes for Health Research (MOP74667) and the Natural Sciences and Engineering Research Council of Canada (NSERC 458115/211598) to JPR.

## Additional information

### Funding

| Funder | Grant reference number | Author |
| --- | --- | --- |
| Canadian Institutes of Health Research | MOP74667 | Jonathan P Rast |
| Natural Sciences and Engineering Research Council of | 312221 | Jonathan P Rast |

Canada

The funders had no role in study design, data collection and interpretation, or the decision to submit the work for publication.

## Author contributions

KMB, Conceptualization, Data curation, Formal analysis, Investigation, Writing—original draft, Writing—review and editing; ECHH, Data curation, Formal analysis, Investigation, Writing—original draft; TH, Formal analysis, Investigation, Methodology, Writing—original draft; CSS, Formal analysis, Investigation, Visualization; NWS, Formal analysis, Investigation, Visualization, Methodology; GW, Data curation, Investigation, Visualization; JPR, Conceptualization, Formal analysis, Supervision, Funding acquisition, Writing—original draft, Project administration, Writing—review and editing

## Author ORCIDs

Katherine M Buckley, http://orcid.org/0000-0002-6585-8943
Jonathan P Rast, http://orcid.org/0000-0001-6494-0403

## Ethics

Animal experimentation: All animal care and use protocols were approved by the Sunnybrook Research Institute Animal Care Committee

## Additional files

### Supplementary files

• Supplementary file 1. Sequences of oligonucleotides used for qPCR, RACE, WMISH, and reporter BAC constructs.

• Supplementary file 2. Genomic coordinates for the echinoderm *IL17* genes.

• Supplementary file 3. SEFIR domain-containing proteins.

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
