## [Decision Letter]

Thank you for submitting your article "IL17 factors are early regulators in the gut epithelium during inflammatory response to Vibrio in the sea urchin larva" for consideration by *eLife*. Your article has been favorably evaluated by Wendy Garrett (Senior Editor) and three reviewers, one of whom is a guest Reviewing Editor. The following individual involved in review of your submission has agreed to reveal his identity: Anthony De Tomaso (Reviewer #3).

The reviewers have discussed the reviews with one another and the Reviewing Editor has drafted this decision to help you prepare a revised submission.

In this manuscript by Jonathan Rast's group, the authors describe a detailed analysis of immunity in the purple sea urchin (*Strongylocentrotus purpuratus*). Very little is known about immune responses in this organism, especially at mucosal sites, and the authors use elegant and in-depth evolutionary genomics approaches to identify and classify genes induced during an immune response. Here, the authors describe development of a model for gut-associated immunity in which they infect larvae with Gram-negative bacteria that are trophic for sea urchins (*Vibrio diazotrophicus*, Vd). In identifying genes induced during the early response to infection, they identify a striking upregulation of cytokine-like genes with homology to IL-17 family members. These are analysed temporally and spatially within the organism, and using BAC clones with GFP reporter genes they are able to track protein expression in vivo during infections as well. The authors classify these genes not only within sea urchins, but other types of marine animals, and present this data within the overall evolutionary context. They also identify IL-17 receptor-like molecules, and show that blockade with morpholinos limits immune responses.

Essential revisions:

A concern was raised that the infection experiments have not been sufficiently replicated to allow robust conclusions to be drawn about the temporal regulation of the IL17 genes. For their work with larvae, the authors used pools of larvae, which does address genetic variation between individuals. However, none of the infection experiments themselves are replicated, or at least not as indicated in the text. With respect to adult urchins, data for only three adults is presented for SpIL17-9 (two infected and one sham), and there is considerable variation in the dynamics of the two treatments. Again, this complicates interpretation of the results. In the morpholino experiments, the translational MASO is missing from Figure 7, despite being referred to in the text and figure legend. In Figure 7—figure supplement 2 panel E, the data are normalized twice and result in only modest differences in expression, making it difficult to draw conclusions. In Figure 7—figure supplement 3, it appears that the translational MASO has no effect on IL17 expression, but there is an effect on several downstream signaling molecules. How can this be explained? Also in this figure, why are some MASOs missing from some panels? In several places the authors referred to data in the text that doesn't appear in the paper. See, for example, subsection “A third SpIL17 subfamily is upregulated in adult immune cells”, first paragraph (RNA-seq analysis), subsection “Interfering with IL17 signaling affects downstream gene expression”, third paragraph (analysis of publicly available data), subsection “IL17 Receptors”, first paragraph (missing in situ data).

In a revision, the authors need to (1) clarify the measures used to ensure reproducibility, (2) ensure that all data presented in the paper is in the correct place in the text.

---

## [Author Response]

*A concern was raised that the infection experiments have not been sufficiently replicated to allow robust conclusions to be drawn about the temporal regulation of the IL17 genes. For their work with larvae, the authors used pools of larvae, which does address genetic variation between individuals. However, none of the infection experiments themselves are replicated, or at least not as indicated in the text.*

We appreciate that we did not sufficiently describe the extent to which the infection experiments have been replicated. We have been characterizing this *Vibrio-*based infection model for many years and have measured changes in cell behavior and gene expression for countless batches of larvae. These results are described in part in Ho, et al., Immunology and Cell Biology, 2016. The kinetics of *IL17* gene expression in response to bacterial challenge have been measured in numerous independent infection experiments and assessed qualitatively in many separate in situhybridization experiments. In addition, these data are consistent with results of analyses of IL17 fluorescent protein reporter transgenes (Figure 5—figure supplement 1). We present here representative results from two independent infection experiments (the RNA-Seq data [Figure 1] and a separate experiment on larvae from an independent mate pair assessed by QPCR [Figure 4]). Consistent quantitative measurements of transgene message prevalence from an additional larval sample are included in Figure 5—figure supplement 1. To clarify the reproducibility of these data, we have added a more detailed explanation to the Results section.

*With respect to adult urchins, data for only three adults is presented for SpIL17-9 (two infected and one sham), and there is considerable variation in the dynamics of the two treatments. Again, this complicates interpretation of the results.*

As outbred animals that exhibit high levels of heterozygosity and immune history, variations in the dynamics of gene expression are common among adult sea urchins responding to immune challenge. We have added data from an additional, independent experiment to the figure to support the reproducibility of these results. Additionally, in this manuscript, the primary conclusions from the adult experiments are that 1) a unique subfamily of *il17* genes (*spil17-9*) is expressed in adult mesodermally derived immune cells, and 2) this expression is not observed in immunoquiescent animals, but is instead dependent on immune stimulation (either bacterial or through injury). The data are consistent on these points and we have clarified the conclusions in the text.

*In the morpholino experiments, the translational MASO is missing from Figure 7, despite being referred to in the text and figure legend.*

We have removed this reference from the text and figure legend, and instead refer the readers to Figure 7—figure supplement 3. We focus on the splice blocking MASO as the efficiency of this is directly quantifiable by QPCR and present the parallel results from the translation blocking MASO to demonstrate specificity.

*In Figure 7—figure supplement 2 panel E, the data are normalized twice and result in only modest differences in expression, making it difficult to draw conclusions.*

We agree. The figure was intended to show that there may be a moderate differential change in the expression levels of the two splice forms over the course of infection. However, because the measurements are complex and the conclusion is an aside in the paper, we have removed this data and will revisit it in future studies.

*In Figure 7—figure supplement 3, it appears that the translational MASO has no effect on IL17 expression, but there is an effect on several downstream signaling molecules. How can this be explained? Also in this figure, why are some MASOs missing from some panels?*

We have clarified this experiment in the text to explain the two types of perturbation; the translation MASO does not affect IL17R1 mRNA prevalence (top row, left most panel). Translation blocking MASOs have been used extensively in the sea urchin system by us and many other laboratories (in particular, to generate the data use to construct the endomesodermal gene regulatory network). Except in special cases of regulatory feedback, translation-blocking MASOs most often do not affect mRNA levels of their targets. The results shown are therefore consistent with many translation blocking MASOs. The splice blocking MASO specifically interferes with splicing of exons 15 and 16 (we have added a sequence analysis to this effect to in Figure 7—figure supplement 2. This can be assessed directly by qPCR and compared to other exons junctions (e.g., exons 3 and 4) to demonstrate MASO efficacy.

*In several places the authors referred to data in the text that doesn't appear in the paper. See, for example, subsection “A third SpIL17 subfamily is upregulated in adult immune cells”, first paragraph (RNA-seq analysis), subsection “Interfering with IL17 signaling affects downstream gene expression”, third paragraph (analysis of publicly available data), subsection “IL17 Receptors”, first paragraph (missing in situ data).*

We have added the RNA-Seq data generated from larvae responding to *Vibrio diazotrophicus* infection to Genbank (BioProject PRJNA380184). Additionally, the in situhybridization for *SpIL17-R1* is now shown in Figure 7—figure supplement 2.